# Description of *Bothriurus mistral* n. sp., the highest-dwelling *Bothriurus* from the western Andes (Scorpiones, Bothriuridae), using multiple morphometric approaches

**Andrés A. Ojanguren-Affilastro**[1]*, **Hugo A. Benítez**[2,3], **Hernán A. Iuri**[1], **Camilo I. Mattoni**[4], **Fermín M. Alfaro**[5,6,7], **Jaime Pizarro-Araya**[5,7,8]

**1** División Aracnología, Museo Argentino de Ciencias Naturales Bernardino Rivadavia (CONICET), Buenos Aires, Argentina, **2** Laboratorio de Ecología y Morfometría Evolutiva, Centro de Investigación de Estudios Avanzados del Maule, Instituto Milenio Biodiversidad de Ecosistemas Antárticos y Subantárticos (BASE), Universidad Católica del Maule, Talca, Chile, **3** Centro de Investigación en Recursos Naturales y Sustentabilidad (CIRENYS), Universidad Bernardo O'Higgins, Santiago, Chile, **4** Laboratorio de Biología Reproductiva y Evolución, Instituto de Diversidad y Ecología Animal (IDEA, CONICET–UNC), Facultad de Ciencias Exactas, Físicas y Naturales, Universidad Nacional de Córdoba, Córdoba, Argentina, **5** Laboratorio de Entomología Ecológica (LEULS), Departamento de Biología, Facultad de Ciencias, Universidad de La Serena, La Serena, Chile, **6** Instituto de Investigación Multidisciplinar en Ciencia y Tecnología, Universidad de La Serena, La Serena, Chile, **7** Grupo de Artrópodos, Sistema Integrado de Monitoreo y Evaluación de Ecosistemas Forestales Nativos (SIMEF), Santiago, Chile, **8** Instituto de Ecología y Biodiversidad (IEB), Santiago, Chile

* andres.ojanguren@gmail.com, ojanguren@macn.gov.ar

## Abstract

We describe *Bothriurus mistral* n. sp. (Scorpiones, Bothriuridae) from the Chilean north-central Andes of the Coquimbo Region. This is the highest elevational discovery for *Bothriurus* in the western slopes of the Andes. This species was collected in the Estero Derecho Private Protected Area and Natural Sanctuary as part of the First National Biodiversity Inventory of Chile of the Integrated System for Monitoring and Evaluation of Native Forest Ecosystems (SIMEF). *Bothriurus mistral* n. sp. is closely related to *Bothriurus coriaceus* Pocock, 1893, from the lowlands of central Chile. This integrative research includes a combination of traditional morphometrics and geometric morphometric analyses to support the taxonomic delimitation of the species.

## Introduction

The genus *Bothriurus* is the most widespread taxon among Bothriuridae, occupying most environments of southern South America, from the arid cold areas of southern Patagonia to the tropical rainforests of Amazonia [1]. Despite its high diversity and extended distribution, its presence in high altitudes of the Andes is comparatively marginal [2]. As of now, only three species have been recorded in this environment: *Bothriurus bocki* Kraepelin, 1911 and *Bothriurus trivittatus* Werner, 1939, both from Bolivia and belonging to the *inermis* group

**Data Availability Statement:** All relevant data are within the manuscript.

**Funding:** DIDULS PR2121210 and DIDULS PEQMEN212124 projects of the University of La Serena, Chile, and the funding from the Ministry of Education of Chile, through MINEDUC's performance agreement: Implementation of a competitive model of innovation and creation: preparing the University of La Serena for 2030, ULS19101, ANID FB210006 grant, and the SIMEF project (INFOR-IEB Agreement) to JPA. PICT 2019-597 project by Agencia Nacional de Promoción Científica y Tecnológica (Argentina) to AAOA.

**Competing interests:** The authors have declared that no competing interests exist.

[3, 4], and *Bothriurus olaen* Acosta, 1998, from central Argentina and southern Bolivia, belonging to the *burmeisteri* group [5, 6]. All these species occur in the eastern slopes of the Andes, and no species of this genus has ever been collected on the western slopes of this mountain range. This absence is particularly intriguing because several Chilean species of *Bothriurus* occupy arid hilly areas that are environmentally similar to the arid, high Andean slopes [7–11].

Over the last years our work group has been recording information about the arthropod fauna of Chile via a systematic survey of the whole biota of the country conducted as part of the Integrated System for Monitoring and Evaluation of Native Forest Ecosystems (SIMEF by its Spanish acronym). As a part of this project we have recently performed the first arthropod survey in the protected high-Andean area of the Estero Derecho Private Protected Area and Natural Sanctuary, in the Coquimbo region, central Chile. As a result, our group has managed to collect several species of scorpions, two of which were unknown for the area, including the first high-altitude Andean species of *Bothriurus* ever collected in the western slopes of the Andes.

In this article a combination of methodologies were used to describe *Bothriurus mistral* n. sp. from the Chilean central Andes. This species is most closely related to the lowland species *Bothriurus coriaceus* Pocock, 1893. Herein we combined traditional taxonomy and geometric morphometrics to provide a better understanding of the morphological trait variation between species, and better support the taxonomic delimitation of the new species.

The presence of this new species in the Andean area of Coquimbo region is remarkable not only because this is the first record of this genus for western Andes, but also because this area has been extensively surveyed in the last two decades [11–14], with not a single previous record of this genus in the area [11].

## Materials and methods

### Study area

The fieldwork was conducted in the Estero Derecho Private Protected Area and Natural Sanctuary, located in the transverse valley range in the northernmost part of the Coquimbo Region, in the Andean area of the Elqui valley (Coquimbo Region, Chile) (Figs 1 and 2B and 2C). The area is located 125 km away from La Serena, upstream from the town of Alcohuaz, and 15 km away from the town of Pisco Elqui [15]. The area is a basin head that provides water from the accumulated winter snow and from multiple rocky glaciers [16]. The large number of high-Andean water meadows present in the area provide water regulation through purification. In addition, the vegetation coverage helps to regulate global-scale climate, temperature and precipitation, as well as erosion, holding the soil in place and preventing landslides. They also harbor a high-Andean biota in good conservation status [15]. In the Andean and high mountainous foothills, the predominant soils are poorly developed entisols and aridisols, which are more common in the steep slopes of rocky hills with little vegetation coverage [17, 18]. The precipitation in the area is mostly nival, with a yearly average of 200 mm and high interannual variability [18]; between 2,000 masl and 3,000 masl, the climate type is cold mountain steppe, which is characterized by strong winds, high solar radiation, and higher winter precipitation, especially nival.

Specimens where collected at night using with UV lights. Each collection site in the study area was defined as "Conglomerate" following current standards of the SIMEF. Photos of environment and living specimens in the field (Fig 2) were taken by our group using digital cameras or drones with attached digital cameras.

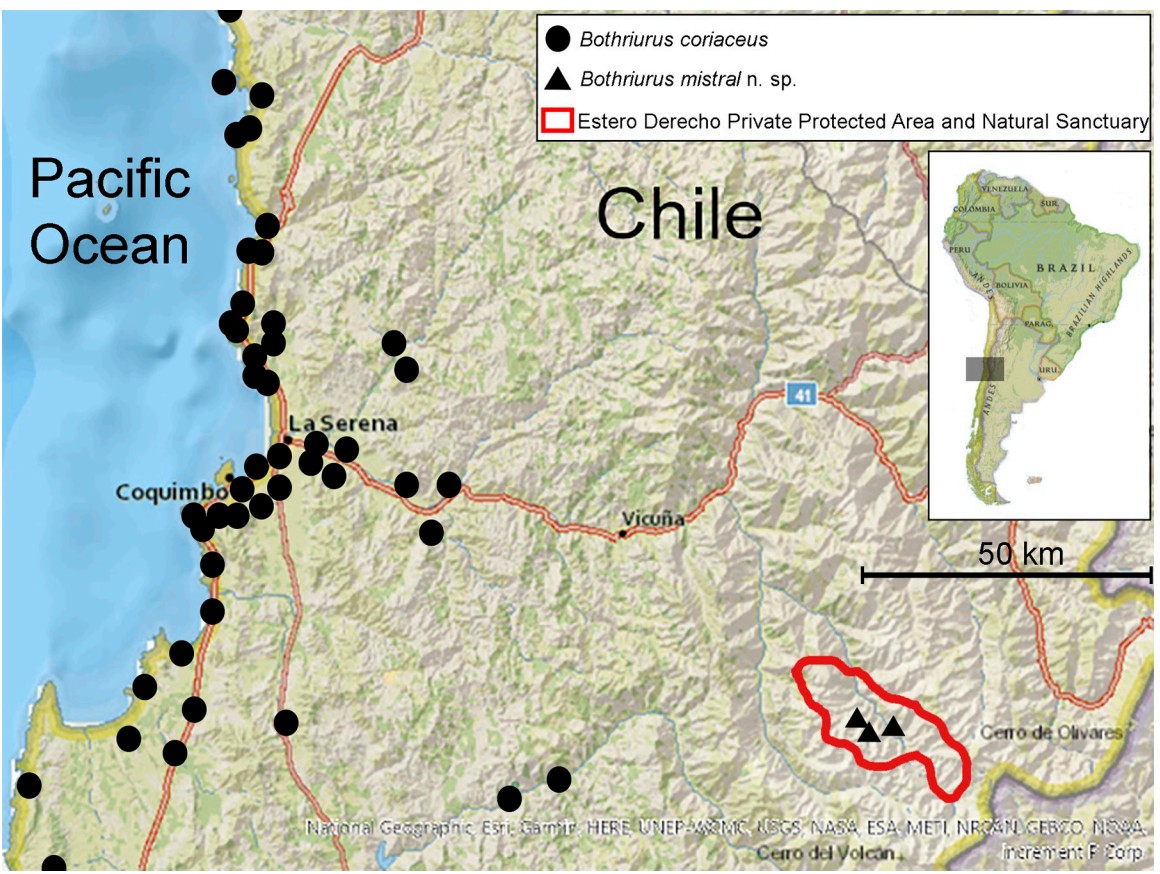

**Fig 1. Distribution map of *Bothriurus mistral* n. sp. (black triangles) and distribution of *Bothriurus coriaceus* in neighboring areas (black circles).** Estero Derecho Private Protected Area and Natural Sanctuary marked with a red line, (Coquimbo Region, Chile).

## Studied material

Studied and measured specimens of *Bothriurus mistral* n. sp. correspond to the type series (see below).

Studied and measured specimens of *Bothriurus coriaceus*: Cuesta Porotitos, [29˚45′42.61″S, 71˚18′14.19″W], Elqui Province, Coquimbo Region, 26/X/2011, 1 male, Ojanguren-Affilastro & Pizarro-Araya coll.; 30 km N. Ovalle, [30˚22′37.68″S, 71˚13′55.17″W], Limarí Province, Coquimbo Region, 10/I/1984, 1 male, Maury coll.; Los Vilos, [31˚55′17.88″S, 71˚30′52.27″W], Choapa Province, Coquimbo Region, 30/IX/1983, 2 males, E. Mary coll.; Quebrada Playa Agua Dulce, 45 km N. Los Vilos, [31˚29′25.84″S, 71˚33′41.82″W], Choapa Province, Coquimbo Region, 5–6/XI/1988, 3 males, Maury coll.; same locality and collector, 8/I/1984, 1 male; Juncal, [32˚52′34.08″S, 70˚12′21.39″W], (1,980 masl), Aconcagua Province, Valparaíso Region, 5/I/1984, 2 males, Maury coll.; Lampa, [33˚17′42.86″S, 70˚53′37.65″W], Metropolitan Region of Santiago, 19/IX/1977, 2 males, Peña coll.; Guayacán, [33˚36′56.47″S, 70˚20′50.58″W], San José de Maipo, Cordillera Province, Metropolitan Region of Santiago, 5/I/1984, 3 males, Maury coll.

## Taxonomic methodology

The descriptive terminology follows Maury [19] for the hemispermatophores; Vachon [20] for the trichobothria; Loria & Prendini [21] for the lateral eyes; Roig-Alsina & Maury [22] for the

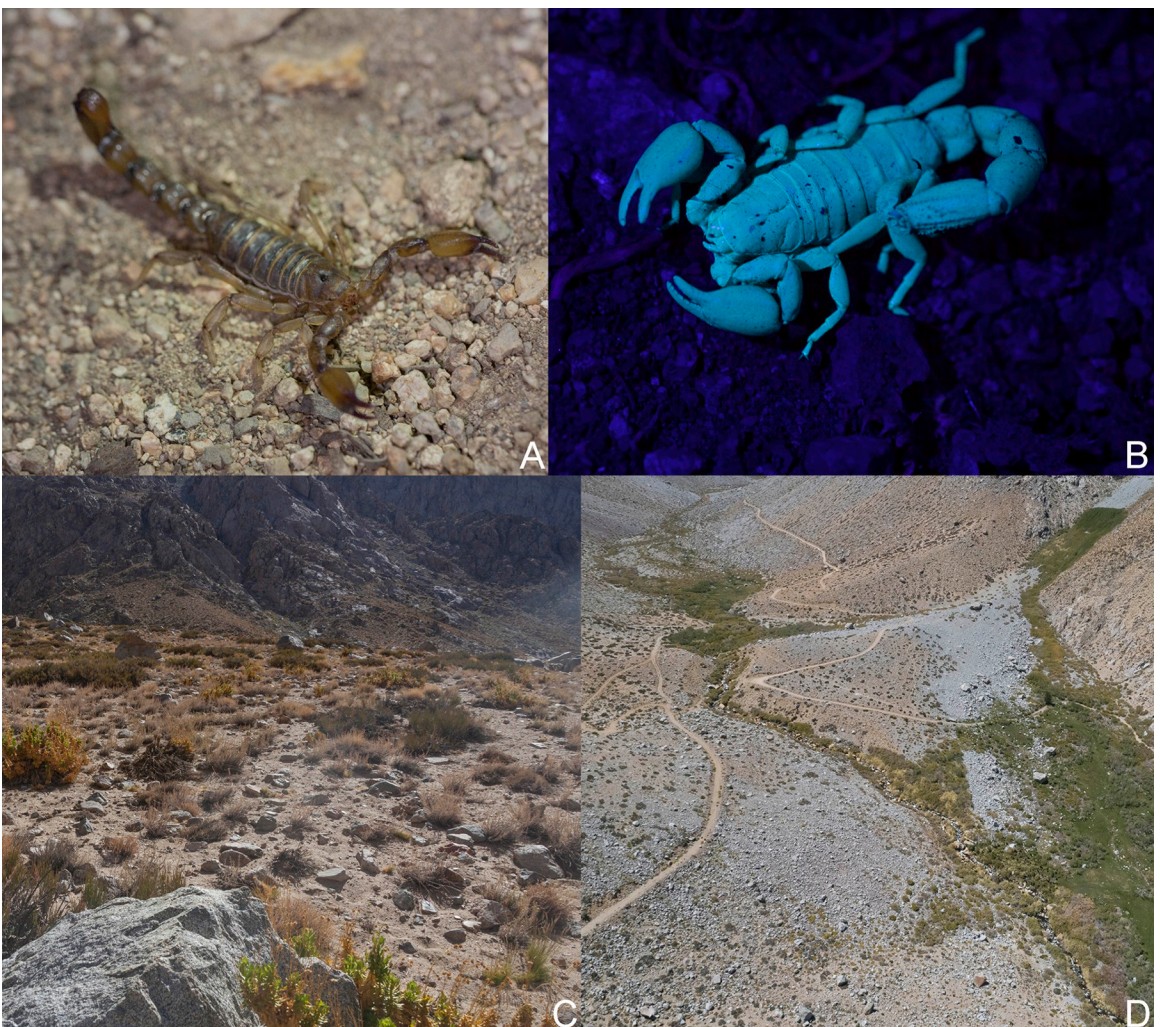

**Fig 2.** (A–D) Living specimens and type locality of *Bothriurus mistral* n. sp. herein described. (A) *Bothriurus mistral* n. sp. (B) *Bothriurus mistral* n. sp. under UV light. (C) Type locality of *Bothriurus mistral* n. sp. in the Estero Derecho Private Protected Area and Natural Sanctuary (Coquimbo Region, Chile). (D) Aerial view of the type locality.

male telson gland; and Ochoa et al. [23] for the metasomal carinae, which are abbreviated as follows: DL = dorsolateral, LIM = lateral inframedian, LM = lateral median, VL = ventrolateral, VM = ventromedian. The terminology for pedipalp carinae follows Prendini [24] and are abbreviated as follows: DI = dorsal internal, DE = dorsal external, VI = ventral internal, VE = ventral external. The abbreviations for collections are as follows: LEULS: Laboratorio de Entomología Ecológica, Universidad de La Serena, La Serena, Chile (Jaime Pizarro-Araya); MACN: Museo Argentino de Ciencias Naturales "Bernardino Rivadavia", Buenos Aires, Argentina (Martín Ramírez); MZUC: Museo de Zoología de la Universidad de Concepción, Concepción, Chile (Jorge Artigas); MNHN: Museo Nacional de Historia Natural, Santiago, Chile (Mario Elgueta). All collection sites were georeferenced using a GPS (Garmin® Etrex model). A distribution map (Fig 1) of *Bothriurus mistral* n. sp. and neighboring records of *B. coriaceus* was generated using USGS National Map Viewer (http://viewer.nationalmap.gov/viewer/); records of *B. coriaceus* were taken from Mattoni & Acosta [10], and more recent collection records from LEULS. All National and International collection and

transport permits were duly obtained, including those provided by the local community managing the Private Protected Area and Natural Sanctuary of Estero Derecho.

## Nomenclatorial acts

The electronic edition of this article conforms to the requirements of the amended International Code of Zoological Nomenclature, and hence the new names contained herein are available under that Code from the electronic edition of this article. This published work and the nomenclatural acts it contains have been registered in ZooBank, the online registration system for the ICZN. The ZooBank LSIDs (Life Science Identifiers) can be resolved and the associated information viewed through any standard web browser by appending the LSID to the prefix "http://zoobank.org/". The LSID for this publication is: urn:lsid:zoobank.org:pub: 248BD623-4F1A-4676-890F-55A6D9C6A408. The electronic edition of this work was published in a journal with an ISSN, and has been archived and is available from the following digital repositories: PubMed Central, LOCKSS and CONICET (Consejo Nacional de Investigaciones Científicas y Técnicas, Argentina).

## Microscopy and images

The measurements, taken using an ocular micrometer, were recorded in millimeters. The hemispermatophores were dissected and then examined in 80% ethanol and photographed under UV and white light. Digital images of pigmentation pattern and habitus were taken under visible light and images of the external morphology under UV light, using a Leica DFC290 digital camera attached to a Leica M165C stereomicroscope, and the focal planes fused with Helicon Focus 3.10.3 (http://helicon.com.usa/heliconfocus/).

## Geometric morphometric approach

In order to have a better understanding of the variation between species, a geometric morphometrics approach was performed using landmarks in the metasomal segment V of the males. We used only males due to a lack of females, and because in this genus it is very difficult to clearly separate adult females from subadults. We chose this morphological trait as a trait with significant taxonomical differences between the species. A total of twenty landmarks were digitized (Fig 3), using the software TPSDig2 [25].

Landmarks of metasomal segment V were defined as this: 1) right rear end of the anal arch; 2) left rear end of the anal arch; 3) ventro-lateral right macroseta of the anal arch; 4) ventro-lateral left macroseta of the anal arch; 5) right distal end of VL carina; 6) left distal end of VL carina; 7) ventro-median right macroseta of the anal arch; 8) ventro-median left macroseta of the anal arch; 9) posterior margin of the right dorsal distal anterior projection; 10) posterior margin of the left dorsal distal anterior projection; 11) right posterior margin of ventral articulation with metasomal segment IV; 12) left posterior margin of ventral articulation with metasomal segment IV; 13) right anterior lateral seta; 14) left anterior lateral seta; 15) joining point between right VL and VSM carinae; 16) right medial lateral macroseta; 17) extreme right external margin; 18) joining point between left VL and VSM carinae; 19) left medial lateral macroseta; 20) extreme left external margin.

The landmarks were aligned using a Procrustes superimposition analysis, by removing size, position and orientation information to standardize the shape of each specimen based on their centroid size [26]. A measurement error (ME) analysis was performed after digitizing the landmarks twice, this data was compared using a Procrustes ANOVA, in which the mean squares (MS) of error should be smaller than the MS of the individual [27].

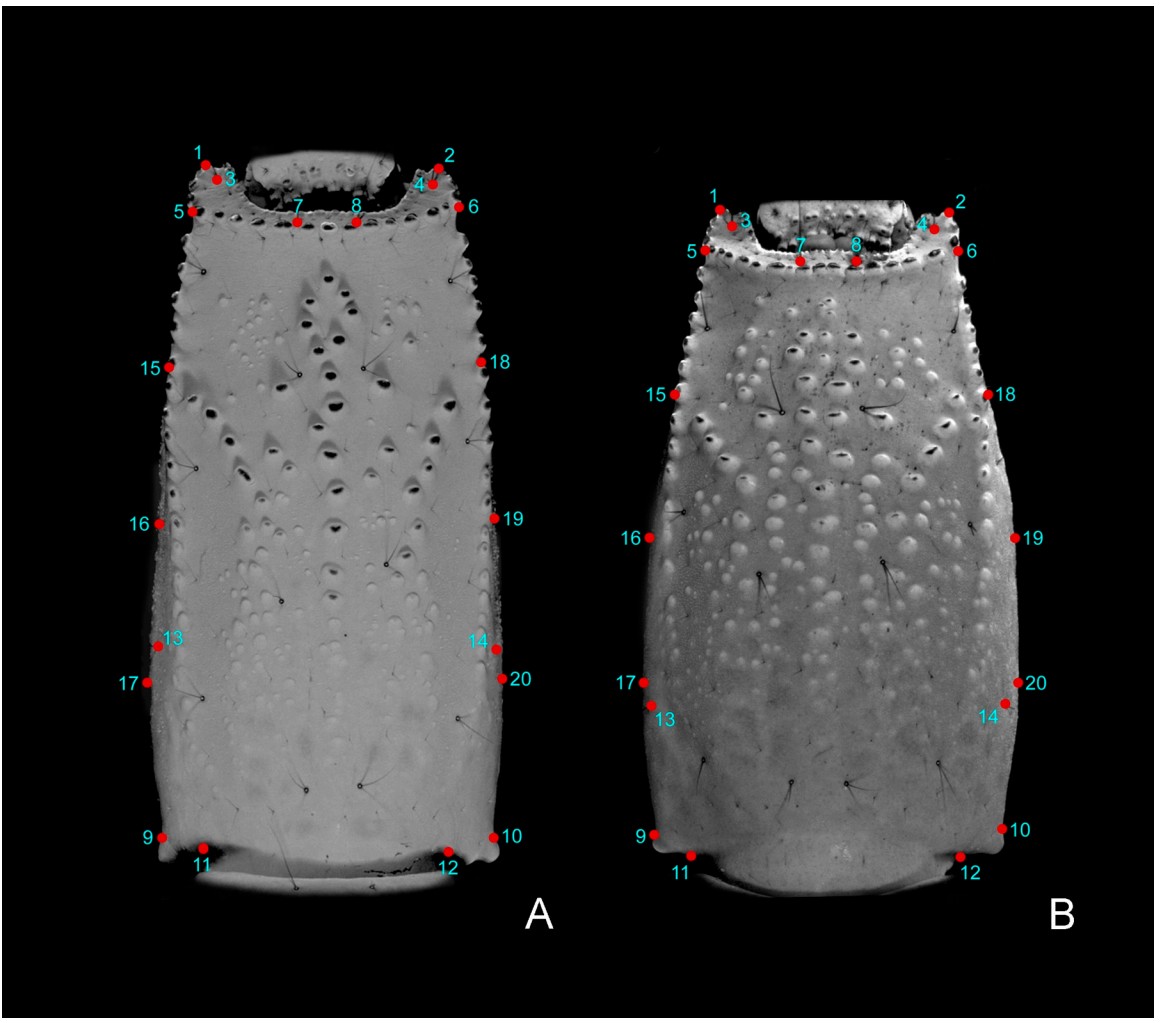

**Fig 3. General description of the location of the 20 landmarks in the ventral surface of metasomal segment V.** (A) *Bothriurus mistral* n. sp. (B) *Bothriurus coriaceus*.

Since the *x* and *y* coordinate were extracted by the Procrustes analysis, the shape variation was analyzed in the entire dataset, using a principal component analysis (PCA) based on the covariance matrix of the individuals' shape. Since shape changes are associated with changes in size, an allometric analysis was performed using a multivariate regression of shape on centroid size [28]. The residuals were used to investigate the shape variation independent of size [29]. To assess whether there are significant differences between the compared species, a random permutation analysis between groups and a Procrustes ANOVA was performed using the RRPP package in R [30].

## Results

### Taxonomy

*Bothriurus mistral* n. sp. Ojanguren-Affilastro, Mattoni, Alfaro & Pizarro-Araya (Figs 1–10 and Table 1) urn:lsid:zoobank.org:act:1024E801-6EC5-43D8-BC49-F7A5BDD667B1.

**Type material.** Holotype male: Chile, Coquimbo Region, Estero Derecho Private Protected Area and Natural Sanctuary, El Chañar Refuge, 30°23′3.86″S, 70°24′44.97″W, 3,034

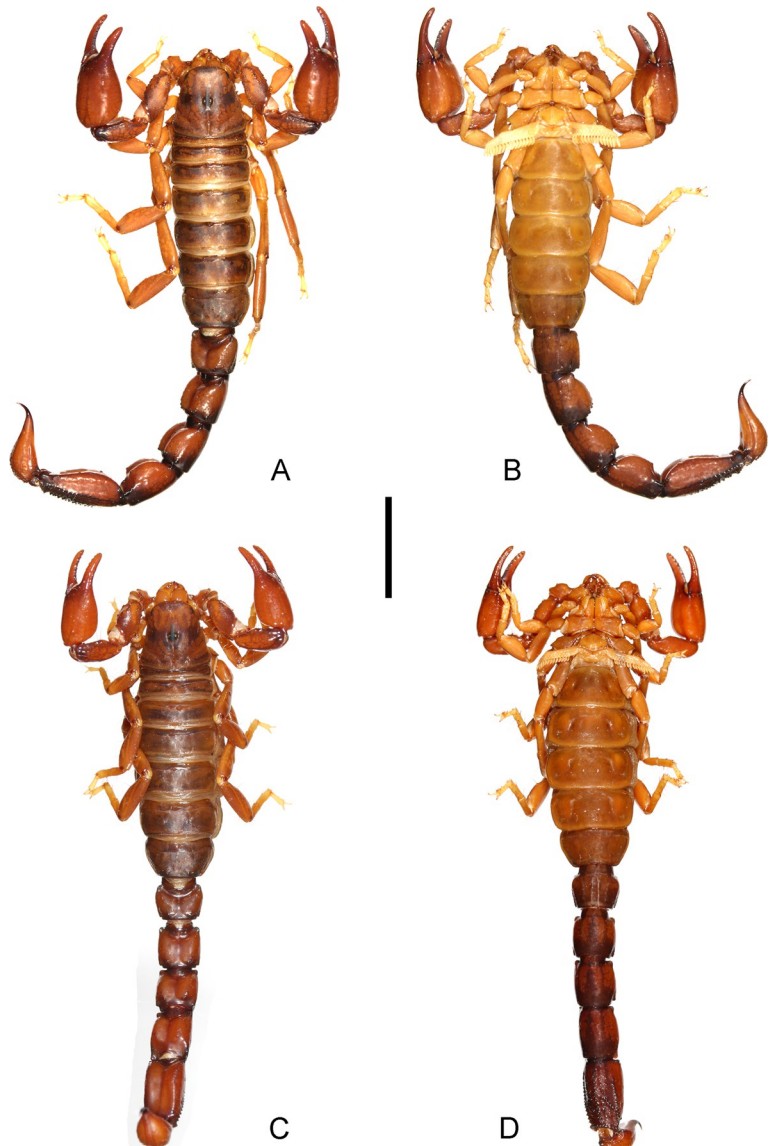

**Fig 4.** (A–D) *Bothriurus mistral* n. sp., habitus. (A) Male, dorsal aspect, (B) Male, ventral aspect, (C) Female, dorsal aspect, (D) Female, ventral aspect. Scale bars: 10 mm.

masl (SIMEF-Project), 24–26/XI/2020, J. Pizarro-Araya, F.M. Alfaro, J. Calderón, A. Castex coll. (MNHN 8376); Paratypes, Conglomerate no. 41141, 30°22′4.78″S, 70°25′51.30″W, 3,015 masl (SIMEF-Project), 24–26/XI/2020, J. Pizarro-Araya, F.M. Alfaro, J. Calderón, A. Castex coll., 9 males, 2 females (MACN); 1 male, 6 females, 2 juveniles (MNHN); 1 male, 3 females, 1 juvenile (MZUC); 1 male, 7 females, 2 juveniles (LEULS), Conglomerate no. 41142, 30°21′58.26″S, 70°22′44.63″W, 3,873 masl (SIMEF-Project), 24–26/XI/2020, J. Pizarro-Araya, F. M. Alfaro, J. Calderón, A. Castex coll., 5 males, 1 female (MACN); 1 male, 4 females, 1 juvenile (MNHN); 2 females, 1 juvenile (MZUC); 1 male, 3 females, 1 juvenile (LEULS).

**Etymology.** The specific name *mistral* is a noun in apposition referring to Gabriela Mistral, pseudonym of the Chilean poetess Lucila María Godoy Alcayaga (1889–1957), who was born in Vicuña and raised in Monte Grande, both in the Elqui valley (Coquimbo Region), an

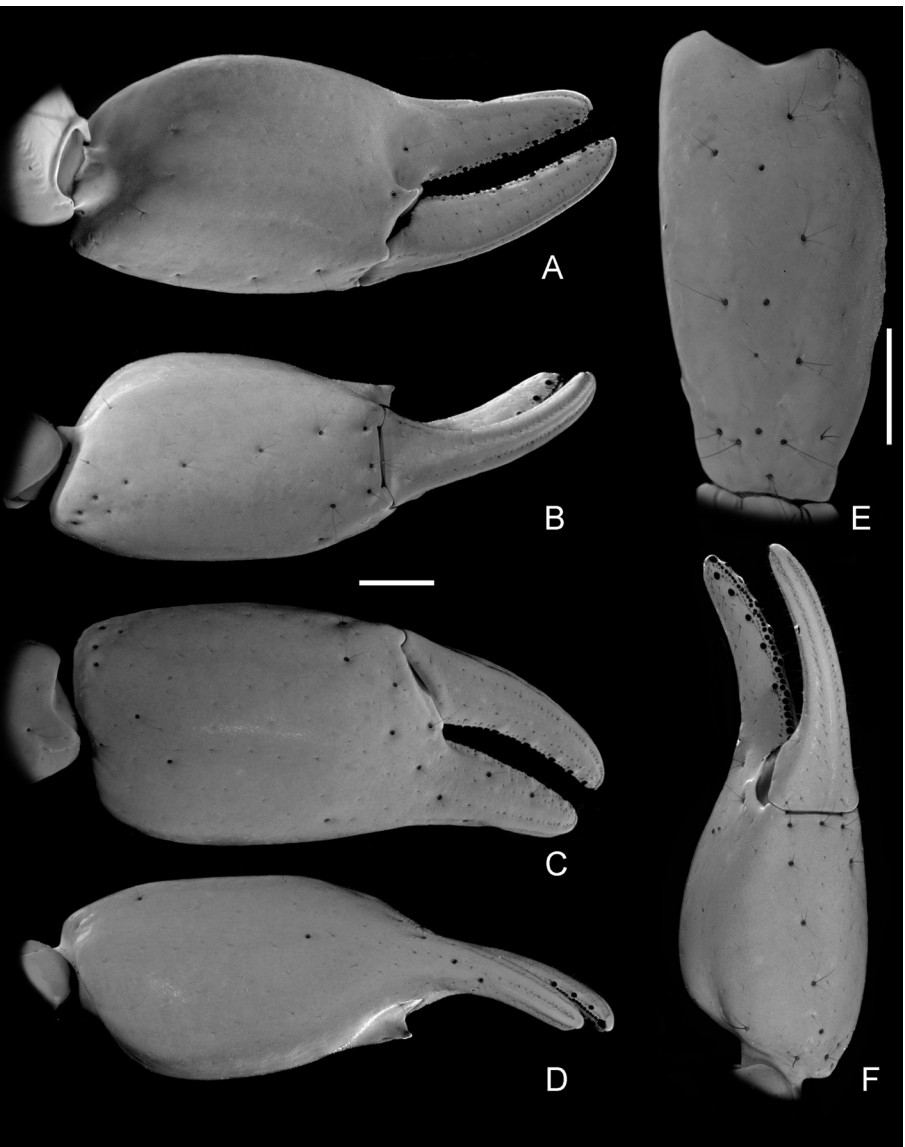

**Fig 5.** (A–F) *Bothriurus mistral* n. sp. (A) Left pedipalp chela, male, internal aspect, (B) Left pedipalp chela, male, ventral aspect, (C) Left pedipalp chela, male, external aspect, (D) Left pedipalp chela, male, dorsal aspect, (E) left pedipalp femur, male, external aspect, (F) Left pedipalp chela, female, ventrointernal aspect. Scale bars: 1 mm.

area adjacent to the type locality of this species. For her poetry work, she was awarded the Nobel Prize in Literature in 1945 and was the first Ibero-American woman and the second Latin American person to receive the Nobel Prize. Gabriela Mistral worked as a teacher at numerous schools of the Elqui valley and became a leading thinker as to the role of public education. She was also involved in the reformation of the Mexican education system, and since the 1920s, led an itinerant life due to her work as a consul and representative for international organizations in the Americas and Europe.

**Description.** Based on the holotype male (MNHN) and paratypes (LEULS, MACN, MNHN, MZUC). Total length, males: 47–57 mm, (N = 10; mean = 52.8 mm); females: 49–56 mm, (N = 3; mean = 51.38 mm). (Measurements of male holotype and male and female paratypes in Table 1).

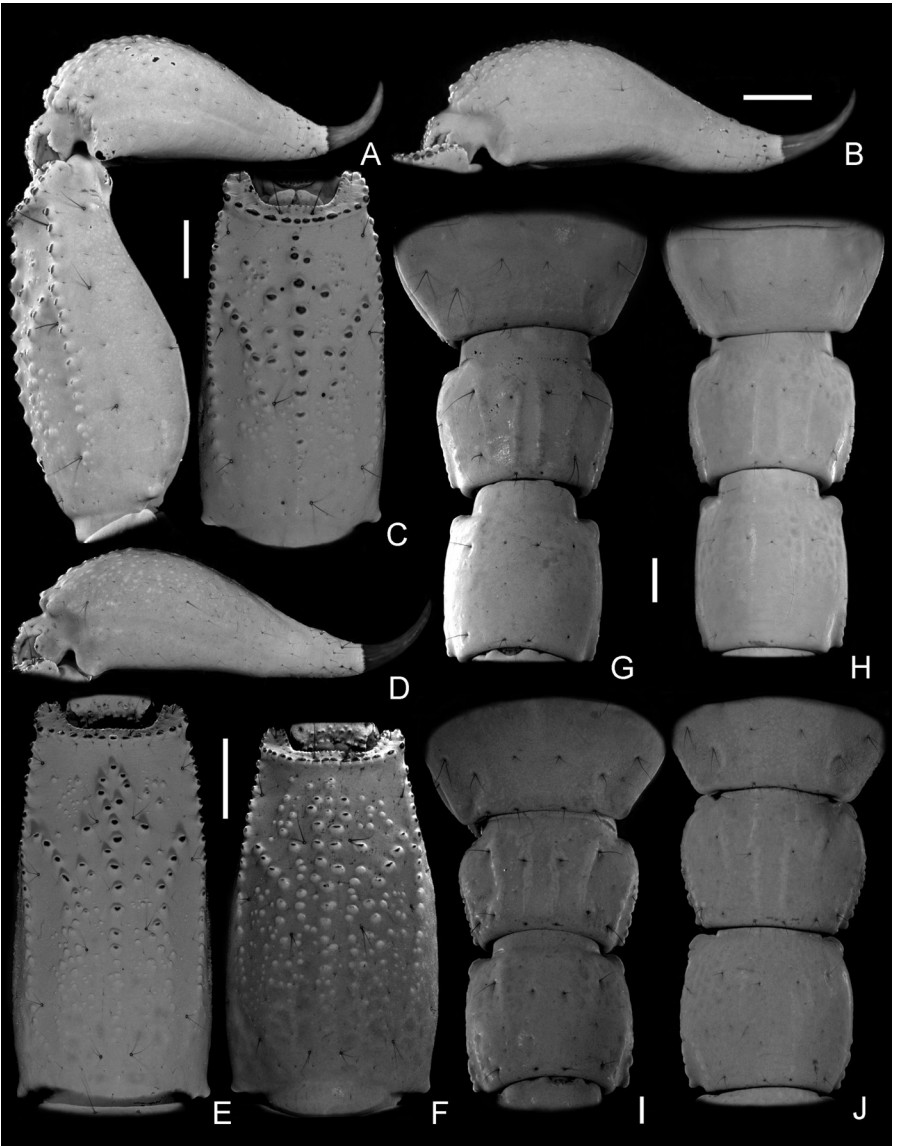

**Fig 6.** (A–J) (A–C, E, G, H). *Bothriurus mistral* n. sp. (A) Telson and metasomal segment V, female, lateral aspect; (B) Telson, male, lateral aspect; (C) Metasomal segment V, female, ventral aspect; (E) Metasomal segment V, male, ventral aspect; (G) Sternite VII and metasomal segments I and II, female, ventral aspect; (H) Sternite VII and metasomal segments I and II, male, ventral aspect; **(D, F–J)** *Bothriurus coriaceus*. (D) Telson, male, lateral aspect; (F) Metasomal segment V, male, ventral aspect; (I) Sternite VII and metasomal segments I and II, female, ventral aspect; (J) Sternite VII and metasomal segments I and II, male, ventral aspect. Scale bars: 1 mm.

*Colour*: Base colour dark yellowish, with dark brown pigmentation pattern in pedipalps, carapace, tergites, metasoma and legs; the remaining yellowish without dark spots (Figs 2A and 4A–4D). Chelicerae with reticular pigmentation on the dorsal margin of manus; external margin of movable finger with a dark spot. Carapace, anterior margin of carapace densely pigmented medially, connecting with two broad dark stripes from the anterior margin to the base of the postocular furrow, lateral margins with an anteromedian wide dark spot, and reticular pigment; posterolateral margins each with a dark spot, surrounded by reticular pigment; ocular tubercle, and area around the lateral eyes dark brown, almost black. Tergites I–VI each

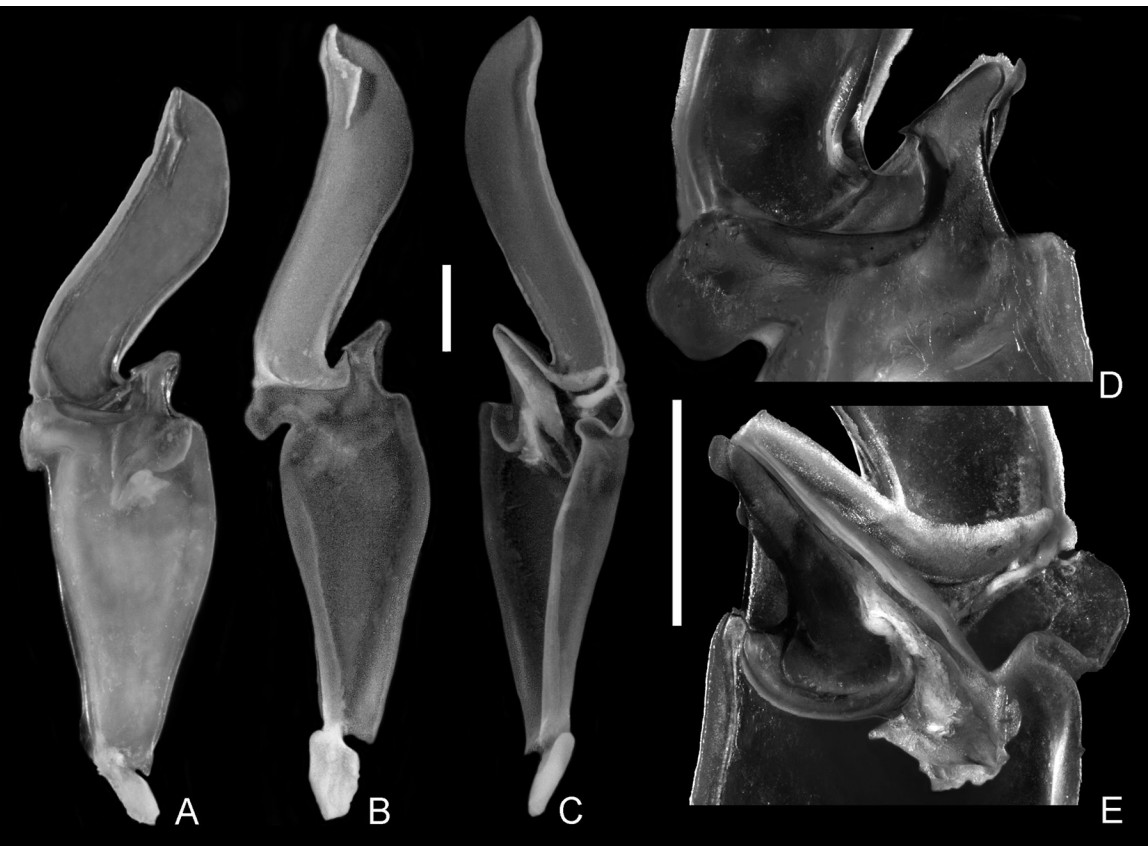

**Fig 7. Hemispermatophore of *Bothriurus coriaceus* and *Bothriurus mistral* n. sp.** (A–E) (A) *Bothriurus coriaceus*, left hemispermatophore, external aspect. (B–E) *Bothriurus mistral* n. sp. Left hemispermatophore (B) External aspect. (C) Internal aspect. (D) Detail of the internal lobe. (E) Detail of lobe region. Scale bars: 1 mm.

with a single transverse dark stripe that covers most part of the anterior margin of each segment, each with internal unpigmented areas on each side; tergite VII similar to segment VI but with a dark spot in the posterior margin and a dark area on the lateral margins. Sternites III–VI with a thin dark stripe on the lateral margin, the rest unpigmented; sternite VII with a thin dark stripe on the lateral margin, medially with faint pigment pattern. Sternum, genital opercula and pectines unpigmented. Metasomal segments I–III: dorsal surface with a dark area in the posterior margin, and a dark area in the articulation; lateral surfaces pigmented in the posterior half of the segment, between LM and LIM carinae, and connecting with ventral pigment; ventral surface with two thin VL dark stripes and a median wide dark stripe, fusing in the posterior half of the segment, but also connected by reticular pattern in the rest of the segment; segment IV similar to segment III but with dorsal surface without the posterior dark area, and with the pigment pattern of lateral margins reduced to a reticulate pigmentation; segment V: dorsal surface with a dark area in the articulation, and faint pigment in DL surface; lateral margins unpigmented; ventral surface with a thin VM stripe, and two wide VL stripes, which become wider towards the posterior margin, fusing in the posterior third of the segment, but connected to each other by reticulate pigment in the anterior two thirds. Telson, vesicle with faint pigment pattern in the ventral margin, the rest unpigmented, dorsal gland of males light yellow; aculeus dark brown. Pedipalps: trochanter with faint reticular pigment pattern; femur with well-developed dark stripes along DI, DE, and VE margins, with DI and DE stripes joining in a big dark dorsal spot occupying almost two thirds of the dorsal surface near

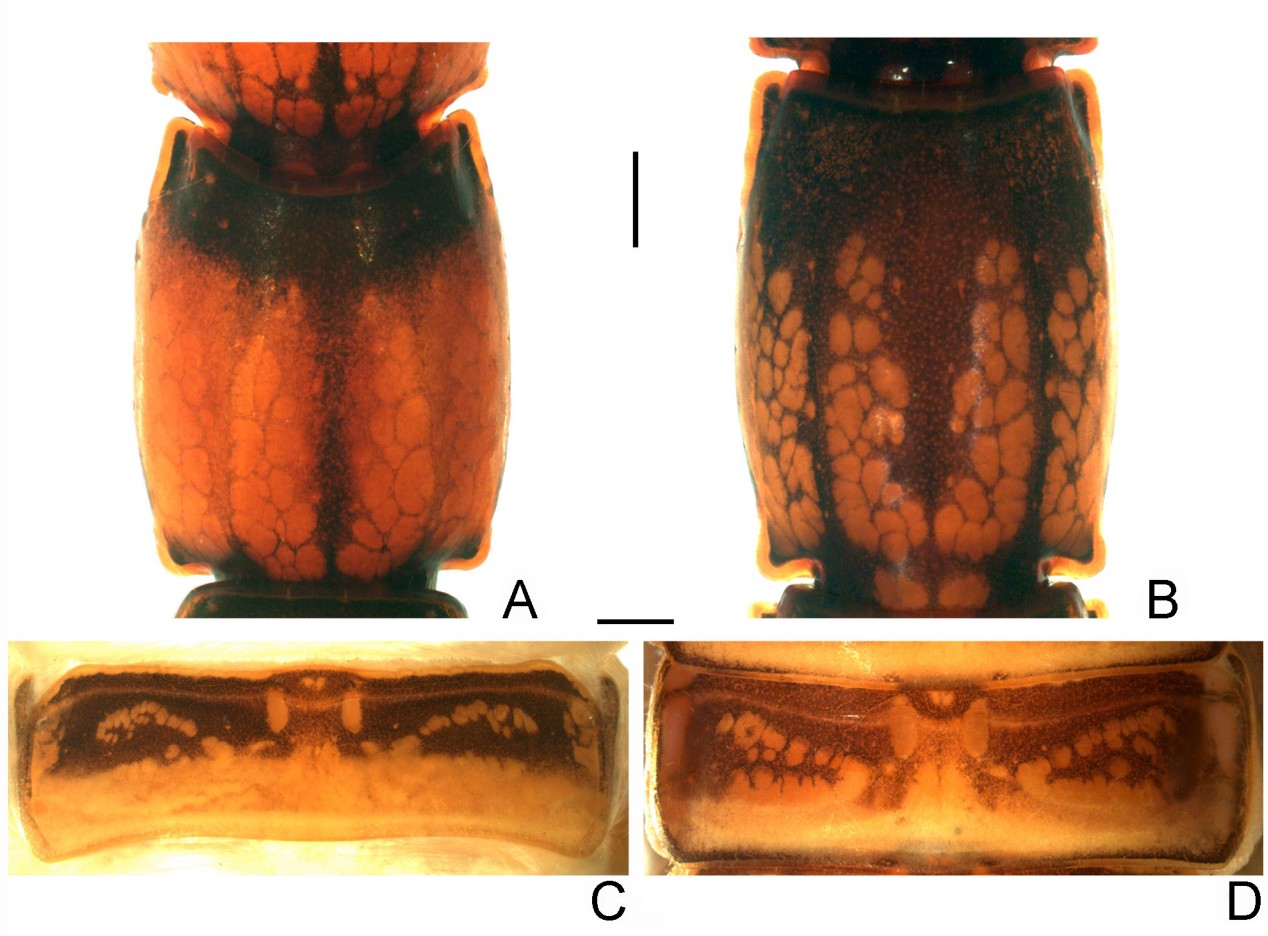

**Fig 8. Diagnostic characters of pigment pattern of *Bothriurus coriaceus* and *Bothriurus mistral* n. sp.** (A, C) *Bothriurus coriaceus*, (A) metasomal segment IV, ventral aspect. (C) terguite IV, dorsal aspect. (B, D) *Bothriurus mistral* n. sp. (B) metasomal segment IV, ventral aspect, (D) terguite IV, dorsal aspect. Scale bars: 1mm.

the articulation with patella. Patella, with dorsointernal and dorsoexternal dark stripes, connected to each other by dense reticulate pattern on the dorsal surface; internal surface densely pigmented in the dorsal half, ventral surface unpigmented. Chela with faint reticular dark stripes along DI, DM, DS, D, E, V and VM carinae; area near articulation of movable finger, and both fingers, with faint pigment. Legs: coxae and trochanters unpigmented; femora faintly pigmented on the external margin and near articulation with patella; patellae pigmented along external and dorsal margins; tibiae with faint pigment pattern along external and dorsal margins, basitarsi, and telotarsi unpigmented.

*Morphology*. *Chelicerae*: anterior margin of movable fingers strongly curved, more so in males; movable fingers with two small subdistal teeth. *Pedipalps*: Femur with DI, DE carinae granular, extending only half of the segment, VI carinae, granular, extending the entire length of the segment; with one DI, one D, and DE macrosetae; anterior margin with few scattered granules; dorsally with some granules near the anterior margin, the rest of the intercarinal surfaces smooth. Patella, DI and VI carinae formed by an elevation of the tegument, remaining surfaces smooth (Fig 5E); with one DI and one VI macrosetae. Chela manus (Fig 5A–5D and 5F) robust, length/height ratio, males: 2.02–2.34 (N = 10; mean = 2.23), females: 2.58–2.71

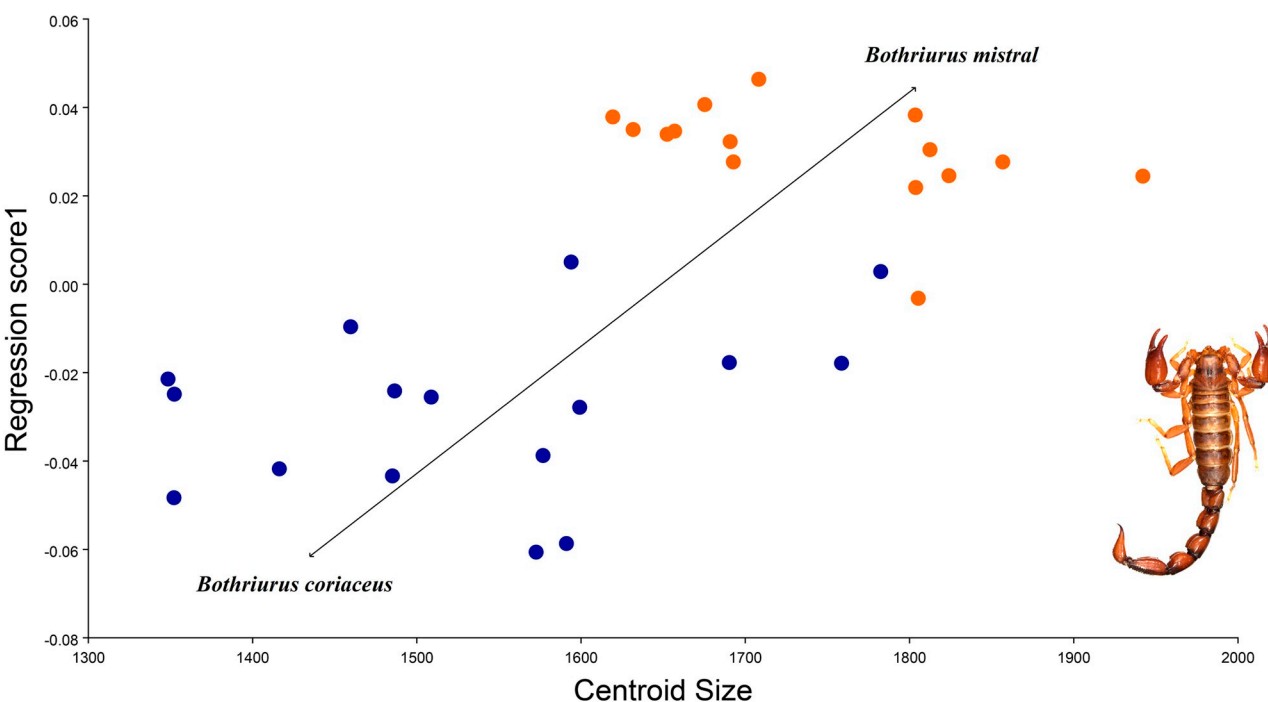

**Fig 9. Multivariate regression of centroid size (independent variable) and shape (dependent variable) between *B. coriaceus* and *B. mistral* n. sp.** The colors represent the different species (blue and orange respectively).

(N = 3; mean = 2.62); length/width ratio, males: 2.47–2.78 (N = 10; mean = 2.64); females: 2.93–3.21 (N = 3; mean = 3.09); internal surface smooth in females (Fig 5F), or with a pronounced, subtriangular projection near articulation of movable finger in males (Fig 5A, 5B and 5D); fingers comparatively short, with a median row of denticles; fixed fingers with six pairs of accessory denticles; movable finger with five external accessory denticles and six internal accessory denticles; the basal external denticle is usually part of the median row. Trichobothrial pattern neobothriotaxic major type C, with one accessory trichobothrium in *V* series of chela; femur with 3 trichobothria (*d*, *i*, *e*), one macroseta (M1) between *d* and *i*; patella with 19 trichobothria (2 *d*, *i*, 3 *et*, *est*, 2 *em*, 2 *esb*, 5 *eb*, 3 *V*), with *esb*2 petite; chela with 27 trichobothria (*Dt*, *Db*, 5 *Et*, *Est*, *Esb*, 3 *Eb*, *dt*, *dst*, *dsb*, *db*, *et*, *est*, *esb*, *eb*, *ib*, *it*, 5 *V*), with *Et*4 petite, *Esb* forming triangle with *Eb*1 and *Eb*2. *Carapace*: anterior margin slightly convex, with two median setae. Surface: finely granular in the dorsolateral margins in males, smooth in females. Anterior longitudinal sulcus absent, postocular furrow present and well developed, lateral sulci shallow. Median ocular tubercle well developed, placed in the middle of the carapace; interocular sulcus barely visible; median ocelli well developed, facing towards the lateral margins, *ca.* two diameters apart, with one seta behind each eye. Lateral ocelli pattern type 3A; with three small lateral ocelli on each side of carapace, two of them anteriorly, third ocellus similar in size to the other two, about one diameter above them. *Legs*: Surfaces smooth. Basitarsi each with two well developed symmetrical pedal spurs. Telotarsi, ventrally with a ventromedian row hyaline setae, and paired ventrosubmedian spines: 1/1 in telotarsus I, 2/2 in telotarsus II and 3/3 in telotarsi III and IV. Ungues curved and short.

*Pectines*: Well developed. Tooth count, males: 15–18 (N = 10; median = 17); females: 13–15 (N = 3; median = 14). *Sternum*: With two conspicuous subtriangular lateral lobes clearly connecting medially, with two macrosetae on each. *Genital opercula*: Sclerites subtriangular. *Tergites*: I–VI: surface smooth; tergite VII with posterior margin finely granular, and four carinae

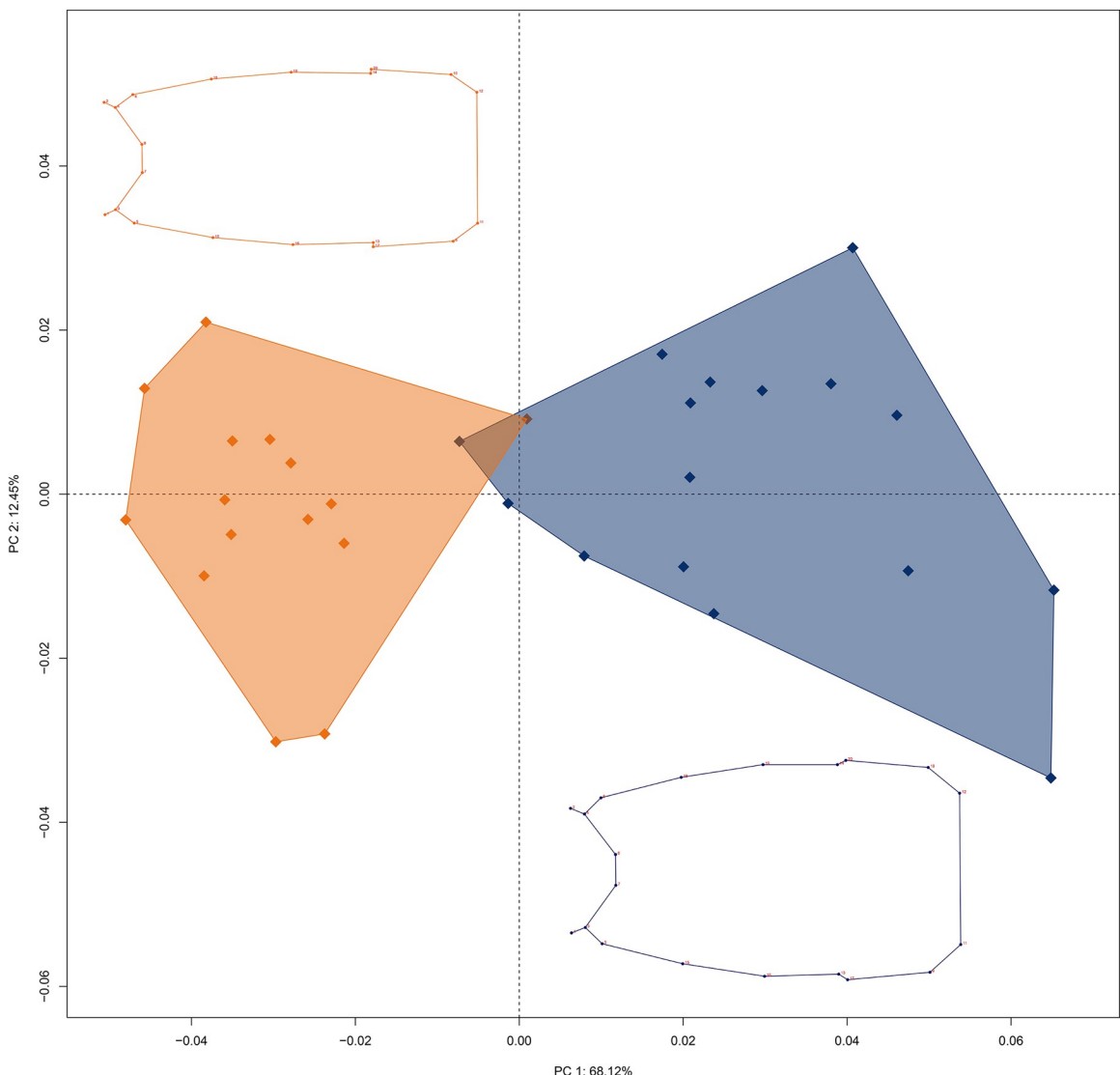

**Fig 10. Principal component analysis of the multivariate regression residuals of the metasomal segment V shape between *B. mistral* n. sp. and *B. coriaceus*, including the average shape wireframe between the species.** The colors of the confidence ellipses represent: orange: *B. mistral* n. sp., blue: *B. coriaceus*.

reduced to a single posterior granule each. *Sternites*: III–VI surface finely granular medially in males, smooth in females, with small elliptical spiracles; sternite VII finely granular medially in males, more densely granular in females, with four poorly developed VSM carinae in females. *Metasoma*: segment I dorsal surface smooth, DL carina formed by scattered granules in the posterior two thirds of the segment, with one DL setae; lateral surface: LSM and LIM carinae restricted to the posterior half of the segment, with some scattered granules between DL and LSM carinae, with one LSM and one LIM setae; ventral surface: females, with four longitudinal carinae occupying the entire length of the segment, two VL and two VSM (Fig 6G); males with smooth tegument (Fig 6H); with eight ventral macrosetae arranged in an anterior row of four macrosetae (two VSM and two VL) and a similar row of four setae in the posterior margin of the segment; segment II similar to segment I, but with a DL macroseta, without LIM macroseta, less granular and with less developed carinae, ventral surface in males and females

**Table 1. Measurements (mm) of *Bothriurus mistral* n. sp.**

| | *Bothriurus mistral* n. sp. | | | | | | |
|---|---|---|---|---|---|---|---|
| | Male holotype | Male paratype | Male paratype | Male paratype | Female paratype | Female paratype | Female paratype |
| Total length | 52.10 | 50.94 | 48.54 | 49.41 | 55.93 | 49.20 | 49.02 |
| Carapace, length | 6.14 | 6.06 | 5.17 | 6.14 | 6.22 | 5.82 | 5.89 |
| Carapace, anterior width | 3.55 | 3.47 | 3.31 | 3.72 | 4.36 | 3.55 | 3.47 |
| Carapace, posterior width | 6.38 | 6.30 | 5.33 | 6.38 | 6.63 | 6.22 | 5.88 |
| Mesosoma, total length | 16.46 | 15.40 | 17.11 | 16.3 | 19.56 | 17.93 | 16.95 |
| Metasoma, total length | 29.50 | 29.48 | 26.26 | 26.97 | 30.15 | 25.45 | 26.18 |
| Metasomal segment I, length | 3.47 | 3.23 | 2.91 | 3.47 | 3.64 | 2.83 | 3.23 |
| Metasomal segment I, width | 4.68 | 4.44 | 3.96 | 4.36 | 3.47 | 4.04 | 3.96 |
| Metasomal segment I, height | 3.45 | 3.23 | 2.99 | 3.55 | 2.67 | 2.99 | 3.07 |
| Metasomal segment II, length | 3.64 | 3.88 | 3.47 | 3.79 | 4.20 | 3.23 | 3.23 |
| Metasomal segment II, width | 4.20 | 4.20 | 3.64 | 4.04 | 3.07 | 3.64 | 3.64 |
| Metasomal segment II, height | 3.39 | 3.39 | 2.99 | 3.55 | 2.66 | 2.99 | 3.07 |
| Metasomal segment III, length | 4.04 | 4.28 | 3.88 | 3.79 | 4.44 | 3.63 | 3.64 |
| Metasomal segment III, width | 4.12 | 4.04 | 3.55 | 3.88 | 2.83 | 3.55 | 3.55 |
| Metasomal segment III, height | 3.47 | 3.55 | 3.15 | 3.47 | 2.59 | 3.15 | 3.15 |
| Metasomal segment IV, length | 4.85 | 4.85 | 4.04 | 4.77 | 4.85 | 4.04 | 4.04 |
| Metasomal segment IV, width | 3.88 | 4.04 | 3.47 | 3.79 | 2.75 | 3.47 | 3.47 |
| Metasomal segment IV, height | 3.39 | 3.31 | 3.15 | 3.47 | 2.34 | 2.99 | 3.23 |
| Metasomal segment V, length | 6.63 | 6.46 | 5.82 | 6.38 | 5.99 | 5.66 | 5.66 |
| Metasomal segment V, width | 3.52 | 3.48 | 3.23 | 3.42 | 3.46 | 3.16 | 3.23 |
| Metasomal segment V, height | 3.15 | 3.23 | 2.83 | 3.15 | 2.10 | 2.91 | 2.91 |
| Telson, length | 6.87 | 6.78 | 6.14 | 6.87 | 7.03 | 6.06 | 6.38 |
| Vesicle, length | 4.85 | 4.68 | 4.2 | 4.77 | 3.79 | 4.44 | 4.69 |
| Vesicle, width | 2.91 | 2.91 | 2.51 | 3.07 | 2.42 | 2.91 | 3.07 |
| Vesicle, height | 2.18 | 2.02 | 1.86 | 2.18 | 2.10 | 2.26 | 2.42 |
| Aculeus, length | 2.02 | 2.10 | 1.94 | 2.10 | 3.24 | 1.62 | 1.70 |
| Chela, length | 9.37 | 8.88 | 8.24 | 9.13 | 8.08 | 8.32 | 8.16 |
| Chela, width | 3.31 | 3.39 | 2.99 | 3.47 | 2.75 | 2.59 | 2.59 |
| Chela, height | 4.04 | 4.12 | 3.55 | 4.12 | 3.12 | 3.07 | 3.15 |

smooth; segment III: as segment II but even less granular and with less developed carinae, DL and LSM carina restricted to some blunt granules in the posterior margin of the segment, LIM carina absent, reduced to a LIM macroseta, ventral surface smooth, the rest as in segment II; segment IV: surfaces smooth, except for some DL granules in males, macrosetae as segment III; segment V: elongated, length/width ratio, males: 1.80–2.01 (N = 10; mean = 1.86); females: 1.73–1.79 (N = 3; mean = 1.75); dorsal surface smooth, medially with a conspicuous furrow, lateral surfaces smooth, with one DL and three LSM macrosetae, ventral surface with granular tegument (Fig 6C and 6E), VL carinae granular, extending the entire length of the segment, with three VL macrosetae (Fig 6A), VSM carinae longitudinal in the anterior half of the segment, diverging in the posterior half of the segment and connecting with the VL carinae in the posterior quarter, with three rows of two VSM macrosetae each, VM carina longitudinal, occupying almost the entire length of the segment; anal arch with four macrosetae. *Telson*: vesicle globose (Fig 6A and 6B), being more elongated in males (Fig 6B), ventral surface granular, dorsal surface smooth, in males there is a median conspicuous dorsal tegumentary depression that corresponds to the telson gland; aculeus short and curved, more elongated in males, base of the aculeus comparatively wide and with a small inconspicuous ventral granule.

*Hemispermatophore*: distal lamina slender, similar in size to the basal portion, slightly curved in its distal third (Fig 7B and 7C); distal crest parallel to the posterior margin, divided by a transversal ridge and occupying less than the distal quarter of the distal lamina; internal lobe with an external apophysis (Fig 7D), we did not observe any small granule nor crest in the depression between the apophysis and the frontal ridge; basal lobe laminar, distally bilobular; frontal ridge well developed; capsular cavity well developed (Fig 7E).

**Diagnosis and comparisons.** *Bothriurus mistral* n. sp. is most closely related to *B. coriaceus*, which occurs in neighboring areas but at much lower altitudes. Both species can be easily told apart by their pigment pattern and morphology. *Bothriurus mistral* n. sp. is more pigmented than *B. coriaceus*; the pigment pattern of tergites I–VI is usually restricted to the anterior third of the segment in *B. coriaceus* (Fig 8C), whereas in *B. mistral* n. sp. it occupies more than the anterior half of each segment (Fig 8D). Additionally, *B. coriaceus* only has a median wide stripe in the ventral surface of metasomal segments I–IV (Fig 8A), whereas *B. mistral* n. sp. has three ventral stripes, two VL and a VM (Fig 8B). *Bothriurus coriaceus* has more developed ventral carinae in metasomal segment I (Fig 6G–6J). Metasomal segment V is more elongated in *B. mistral* n. sp. males (Fig 6E), and its length/width ratio ranges from 1.80 to 2.01 (N = 10; mean = 1.86), whereas in *B. coriaceus* males (Fig 6F) it ranges from 1.61 to 1.76 (N = 10; mean = 1.67). The base of the aculeus in *B. mistral* n. sp. males is slightly thicker and higher than in *B. coriaceus* males (Fig 6B and 6D), but this character is highly variable. There are also some conspicuous differences in the hemispermatophores of both species; in *B. mistral* n. sp. the distal lamina is more elongated and more recurved apically than in *B. coriaceus* (Fig 7A and 7B), being the distal lamina proportionally shorter and wider in *B. coriaceus* than in *B. mistral* n. sp. considering the total length of the hemispermatophore. Hemispermatophore total length/distal lamina length ratio ranges from 1.88 to 1.98 (N = 5; Mean = 1.93) in *B. mistral* n. sp., whereas in *B. coriaceus* it ranges from 2.04 to 2.08 (N = 5; Mean = 2.06). Hemispermatophore total length/distal lamina width ratio ranges from 7.71 to 8.23 (N = 5; Mean = 8.03) in *B. mistral* n. sp., whereas in in *B. coriaceus* it ranges from 6.31 to 7.16 (N = 5; Mean = 6.81).

**Distribution.** *Bothriurus mistral* n. sp. is only known from its type locality, the Estero Derecho Private Protected Area and Natural Sanctuary (30˚23′3.86″S, 70˚24′44.97″W, 3,034 masl), located in the Paihuano mountain range, an area with a surface of 31,680 ha, placed at high altitudes of the north central Chilean Andes (Coquimbo Region, Chile) (Fig 1).

**Ecology.** The area where *Bothriurus mistral* n. sp. was collected occupies the sub-Andean floor, which is characterized by the presence of medium-height shrub vegetation and scrublands, with the dominant species *Stipa chrysophylla* E. Desvaux, 1854, *Viviana marifolia* Cavanilles, 1804, *Cristaria andicola* Gay, 1846, *Adesmia hystrix* Philippi, 1860 (Fabaceae), and *Ephedra americana* Humb. & Bonpl. ex Willd 1806; and the lowermost Andean floor, characterized by a low tropical-Mediterranean shrubland of *Adesmia subterranea* Clos, Gay, C., 1838, and *Adesmia echinus* C. Presl, 1791 (Fabaceae) [15, 31].

*Bothriurus mistral* n. sp. has been found in sympatry with *Brachistosternus gayi* Ojanguren-Affilastro, Pizarro-Araya & Ochoa, 2018, and with an undetermined, possibly new, species of *Brachistosternus*.

## Geometric morphometric analyses

The measurement error was assessed to avoid any digitizing error in the data. The results of a Procrustes ANOVA showed that the MS values of the individuals exceeded the MS values of error, indicating that there is no ME in the data. The PCA between species showed that the first three components explained 85.9% of the variation in shape between *B. coriaceus* and *B. mistral* n. sp. (PC1: 68.1%; PC2: 12.4%; PC3: 5.43%). These results showed that almost all the

variation was concentrated in the first dimension of the shape space and could be a size effect. Therefore, a multivariate regression between shape and centroid size was performed that showed a significant size effect of 25% ($P < 0.001$, after 10,000 permutation rounds) (Fig 9).

After removing all size (allometric) effect, the PCA of the residuals showed a new shape variation explained by the first three components with a 82% (PC1: 59.6%; PC2: 16.1%; PC3: 6.4%) that suggests that the shape variation between *B. coriaceus* and *B. mistral* n. sp. was purely a shape variation (Fig 10).

## Discussion

*Bothriurus mistral* n. sp. belongs to the *coriaceus* group, which is defined in this contribution and also includes *B. coriaceus* from north-central Chile, *Bothriurus keyserlingi* Pocock, 1893 from south-central Chile, and *Bothriurus dumayi* Cekalovic, 1974 from northern Chile [11]. These species occur in lowlands, or at most, at intermediate altitudes, but never above 2,000 masl. [10]. *Bothriurus mistral* n. sp. is the first known high-altitude *Bothriurus* species from the western slopes of the Andes and the first high-altitude species of the *coriaceus* group. This species seems to be very restricted geographically, being only known from its type locality, despite occurring in a relatively well studied area of the Chilean central Andes [12–14]. The lack of previous records of this species could be due to the particular characteristics of the environment where it has been collected, which is an inter-Andean valley surrounding a wetland, a kind of environment usually poorly preserved due to herding, and therefore neglected in arthropod surveys. In this case the exclusion of livestock in the area has preserved an almost pristine environment, representing a rare opportunity to study this kind of Andean wetland. Other neighboring areas with similar environments to the type locality should be surveyed in the future in order to clearly establish the actual distribution of this species. This is a further confirmation that the arthropod fauna of high-altitude environments of the Andes is far more complex and diverse than previously thought.

For this research the inclusion of geometric morphometrics analysis of the metasoma results to be very informative and powerful to discriminate cryptic morphologies [32]. Geometric morphometrics tools have been rarely used in scorpions. Traditionally, some authors used it to evaluate Ecomorphs (morphologies under similar ecological demands) [33, 34], post-embryonic stages [35] or in a paleontological way [36], being this contribution the first to use these techniques to improve a taxonomical description in the group.

This is the first time that a geometric morphometric approach is applied to identify two closely related species of Bothriuridae. Our analyses showed very promising results and these techniques could be implemented for other species of the group in the future, specially to solve the systematic position of taxonomically problematic species.

## Acknowledgments

We are grateful to the community of Estero Derecho Private Protected Area and Natural Sanctuary, and to esteroderecho.cl (Coquimbo Region, Chile) for providing field support and logistics access to their plots. We are also grateful to Juan Calderón (ULS) and Alberto Castex (fundacion-indomita.org) for providing field logistics support, and to Francisco A. Squeo and Lutgarda "Lutty" Arriagada (ULS-IEB) for helping to manage the field work in Estero Derecho Private Protected Area and Natural Sanctuary. We also thank Gerardo Vergara (INFOR) for managing the financing of the project Integrated Forest Ecosystem Assessment and Monitoring System (SIMEF by its Spanish acronym).

## Author Contributions

**Conceptualization:** Andrés A. Ojanguren-Affilastro, Hugo A. Benítez, Hernán A. Iuri, Camilo I. Mattoni, Fermín M. Alfaro, Jaime Pizarro-Araya.

**Data curation:** Andrés A. Ojanguren-Affilastro, Hugo A. Benítez, Fermín M. Alfaro, Jaime Pizarro-Araya.

**Formal analysis:** Andrés A. Ojanguren-Affilastro, Hugo A. Benítez, Hernán A. Iuri, Camilo I. Mattoni, Fermín M. Alfaro, Jaime Pizarro-Araya.

**Funding acquisition:** Jaime Pizarro-Araya.

**Investigation:** Andrés A. Ojanguren-Affilastro, Hugo A. Benítez, Hernán A. Iuri, Camilo I. Mattoni, Fermín M. Alfaro, Jaime Pizarro-Araya.

**Methodology:** Camilo I. Mattoni, Fermín M. Alfaro, Jaime Pizarro-Araya.

**Project administration:** Andrés A. Ojanguren-Affilastro, Fermín M. Alfaro, Jaime Pizarro-Araya.

**Resources:** Andrés A. Ojanguren-Affilastro, Hugo A. Benítez, Camilo I. Mattoni, Jaime Pizarro-Araya.

**Supervision:** Andrés A. Ojanguren-Affilastro, Camilo I. Mattoni, Jaime Pizarro-Araya.

**Validation:** Andrés A. Ojanguren-Affilastro, Hugo A. Benítez, Hernán A. Iuri, Camilo I. Mattoni, Fermín M. Alfaro, Jaime Pizarro-Araya.

**Visualization:** Andrés A. Ojanguren-Affilastro, Hugo A. Benítez, Hernán A. Iuri, Camilo I. Mattoni, Fermín M. Alfaro, Jaime Pizarro-Araya.

**Writing – original draft:** Andrés A. Ojanguren-Affilastro, Hugo A. Benítez, Hernán A. Iuri, Camilo I. Mattoni, Fermín M. Alfaro, Jaime Pizarro-Araya.

**Writing – review & editing:** Andrés A. Ojanguren-Affilastro, Hugo A. Benítez, Hernán A. Iuri, Camilo I. Mattoni, Fermín M. Alfaro, Jaime Pizarro-Araya.

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
