## [Decision Letter · Decision Letter 0]

17 Nov 2022

PONE-D-22-27368Description of Bothriurus mistral n. sp., the highest-dwelling Bothriurus from the western Andes (Scorpiones, Bothriuridae), using multiple morphometric approachesPLOS ONE

Dear Dr. Ojanguren-Affilastro,

Thank you for submitting your manuscript to PLOS ONE. After careful consideration, we feel that it has merit but does not fully meet PLOS ONE’s publication criteria as it currently stands. Therefore, we invite you to submit a revised version of the manuscript that comprehensively addresses the points raised during the review process.

We look forward to receiving your revised manuscript.

Kind regards,

Michael Schubert

Academic Editor

PLOS ONE

2. Please take this opportunity to be sure you have met all of our guidelines for new species. For proper registration of a new zoological taxon, we require two specific statements to be included in your manuscript.

a) In the Results section, the globally unique identifier (GUID), currently in the form of a Life Science Identifier (LSID), should be listed under the new species name, for example:

Anochetus boltoni Fisher sp. nov. urn:lsid:zoobank.org:act:B6C072CF-1CA6-40C7-8396-534E91EF7FBB

Another LSID for the manuscript itself should also appear within the Nomenclature statement. You will need to contact Zoobank (zoobank.org/About) to obtain a GUID (LSID). You should receive one LSID for your manuscript and a separate, unique LSID for the new species. 

b) Please also insert the following text into the Methods section, in a sub-section to be called "Nomenclatural Acts":

The electronic edition of this article conforms to the requirements of the amended International Code of Zoological Nomenclature, and hence the new names contained herein are available under that Code from the electronic edition of this article. This published work and the nomenclatural acts it contains have been registered in ZooBank, the online registration system for the ICZN. The ZooBank LSIDs (Life Science Identifiers) can be resolved and the associated information viewed through any standard web browser by appending the LSID to the prefix "" ext-link-type="uri" xlink:type="simple">http://zoobank.org/". The LSID for this publication is: urn:lsid:zoobank.org:pub: XXXXXXX. The electronic edition of this work was published in a journal with an ISSN, and has been archived and is available from the following digital repositories: PubMed Central, LOCKSS [author to insert any additional repositories].

All PLOS ONE articles are deposited in PubMed Central and LOCKSS. If your institute, or those of your co-authors, has its own repository, we recommend that you also deposit the published online article there and include the name in your article.

Following a recent ruling by the International Commission on Zoological Nomenclature, electronic journals are now a valid format for publication of new zoological taxa. In order to ensure the valid publication of your new species, please be sure to include the updated version of Nomenclatural Acts (above). A complete explanation of our guidelines for publishing new species can be found on our website: http://www.plosone.org/static/guidelines#zoological.

3. Please expand the acronym “MINEDUC” (as indicated in your financial disclosure) so that it states the name of your funders in full.

"We are grateful to the community of Estero Derecho Private Protected Area and Natural Sanctuary, and to esteroderecho.cl (Coquimbo Region, Chile) for providing field support and logistics access to their plots. We are also grateful to Juan Calderón (ULS) and Alberto Castex (fundacion-indomita.org) for providing field logistics support, and to Francisco A. Squeo and Lutgarda “Lutty” Arriagada (ULS-IEB) for helping to manage the field work in Estero Derecho Private Protected Area and Natural Sanctuary. We also thank Gerardo Vergara (INFOR) for managing the financing of the project Integrated Forest Ecosystem Assessment and Monitoring System (SIMEF by its Spanish acronym). AAOA thanks the PICT 2019-597 project by Agencia Nacional de Promoción Científica y Tecnológica (Argentina). JPA thanks the DIDULS PR2121210 and DIDULS PEQMEN212124 projects of the University of La Serena, Chile, and the funding from the Ministry of Education of Chile, through MINEDUC’s performance agreement: Implementation of a competitive model of innovation and creation: preparing the University of La Serena for 2030, ULS19101, ANID FB210006 grant, and the SIMEF project (INFOR-IEB Agreement)."

 "DIDULS PR2121210 and DIDULS PEQMEN212124 projects of the University of La Serena, Chile, and the funding from the Ministry of Education of Chile, through MINEDUC’s performance agreement: Implementation of a competitive model of innovation and creation: preparing the University of La Serena for 2030, ULS19101, ANID FB210006 grant, and the SIMEF project (INFOR-IEB Agreement) to JPA.

PICT 2019-597 project by Agencia Nacional de Promoción Científica y Tecnológica (Argentina)  to AAOA."

5. We note that Figures 1 and 2 in your submission contain [map/satellite] images which may be copyrighted. All PLOS content is published under the Creative Commons Attribution License (CC BY 4.0), which means that the manuscript, images, and Supporting Information files will be freely available online, and any third party is permitted to access, download, copy, distribute, and use these materials in any way, even commercially, with proper attribution. For these reasons, we cannot publish previously copyrighted maps or satellite images created using proprietary data, such as Google software (Google Maps, Street View, and Earth). For more information, see our copyright guidelines: http://journals.plos.org/plosone/s/licenses-and-copyright.

a) You may seek permission from the original copyright holder of Figures 1 and 2 to publish the content specifically under the CC BY 4.0 license.  

Reviewers' comments:

Reviewer's Responses to Questions

**Comments to the Author**

1. Is the manuscript technically sound, and do the data support the conclusions?

Reviewer #1: Yes

Reviewer #2: Yes

2. Has the statistical analysis been performed appropriately and rigorously? 

Reviewer #1: Yes

Reviewer #2: Yes

3. Have the authors made all data underlying the findings in their manuscript fully available?

Reviewer #1: Yes

Reviewer #2: Yes

4. Is the manuscript presented in an intelligible fashion and written in standard English?

Reviewer #1: Yes

Reviewer #2: Yes

5. Review Comments to the Author

Reviewer #1: The manuscript submitted describes a new cryptic species of Bothriurus from an unexplored locality in the Chilean Andes. The description is comprehensive and includes high-quality pictures, including photos under UV light, and photos of the male hemispermatophore. I have a few suggestions that could be easily fixed. One potential relevant issue is to incorporate an explanation on how the landmarks where defined in the segment V of both species (e.g., using carinae, macrosetae, relative position etc.). This way other authors could replicate the landmarks across additional species in further contributions.

Reviewer #2: The manuscript “Description of Bothriurus mistral n. sp., the highest-dwelling Bothriurus from the western Andes (Scorpiones, Bothriuridae), using multiple morphometric approaches” represents an important and interesting contribution to the knowledge of the diversity of scorpions in the region of the north-central Andes. The study not only provides the highest elevational discovery for Bothriurus in the western slopes of the Andes, but also uses the geometric morphometrics analyses for the first time to delimit a recent species within this order of arachnids. It is a great pity that the authors failed to add at least the basic characteristics and variability of genetic markers to the morphometric analyzes of the studied species (B. mistral n. sp. and B. corciareus). In that case, the results would have an even better informative value.

It is a relatively short manuscript that is carefully prepared. I have only a few comments about it, which can contribute to a better use of the presented data in possible subsequent studies.

The results represent interesting original data. In my opinion, this manuscript should be accepted for publication in Plos One after correction of some details.

Comments/recommendations:

Material and Methods

Studied material

Lines 111-119: Please add GPS coordinates also to the material of Bothriurus coriaceus.

Nomenclatorial acts

Lines 146-156: I do not understand this whole long passage. In my opinion, it is clear that the code must be followed and that the name is registered in ZooBank. This entire paragraph can only be replaced by the specified internet link to the registered name.

Geometric morphometric approach

Line 175: Seventeen landmarks are mentioned (Fig. 3). However, 20 points are shown in the Fig. 3. It is not clear; how did you define landmark 13 and 14. The position is clearly different between species.

Results

Line 203: Specify the registration collection number of holotype.

Lines 205-206: What does it mean: “J. Pizarro-Araya, F.M. Alfaro, J. Calderón, A. Castex (MNHN)”. All mentioned persons are owners of the collection or collected (leg.) the samples? Please, clarify this information (in the whole Type material).

Line 206: What does it mean: “Conglomerate no. 41141“. It seems to like registration collection number. In this case I don’t understand that this material may be located in different collections (SIMEF-Project, MACN, MZUC, LEULS). Please, clarify this information and add the real registration collection numbers.

Lines 229-230: Specify separately the measurements for holotype and paratypes (include separately this information also in Table 1). The values of Total length from the text do not correspond to the data in Table 1.

Line 234: The holotype is only one individual (male in this case). Please correct the legend.

Table 1. – Here is no variability of the measurements. The morphometric information seems to be important for the characteristic of species. Please present the minimal, maximal, and mean values for all measurements. And separately for holotype and all paratypes.

Line 238: Duplication of Line 234.

Line 240-278: More then one page of color description seems to be very precise description of this characteristic. However, the coloration may be variable, and it is difficult to navigate in a long text. Moreover, the Figs 2A and 4A–4D are not informative in this case. The pigment pattern is used as important diagnostic characteristics. It is a reason I fully recommended to add more detailed pictures (photographs or drawing) of described coloration (with visible pattern, not as in Figs 2A and 4A–4D). it would be best to also add comparison with B. coriaceus.

Line 397: Terms “shorter and wider“ are quite confusing. Please add specific values in both species.

Please, control the formal style of Figure references all over the text. It is mix of “Fig” and Fig.” Moreover “Figure” is not abbreviated in the figure legends. I assume that the formatting of the text will be subject to an even more detailed check by the editorial team.

6. PLOS authors have the option to publish the peer review history of their article (what does this mean?). If published, this will include your full peer review and any attached files.

Reviewer #1: **Yes: **Jairo A. Moreno-González

Reviewer #2: No

---

## [Author Response · Author response to Decision Letter 0]

2 Jan 2023

Response to reviewers

We went through all the reviewers and editors comments. We accepted all suggested changes; we included the LSID of the new species, as well as an additional figure (Fig. 8) an expanded table 1, and a new figure 1 using USGS National Map Viewer (http://viewer.nationalmap.gov/viewer/), which is one of the sources suggested by PlosOne. All changes are highlighted with track changes in the “Revised Manuscript with Track Changes”, and an unmarked version of the revised paper is also presented labeled as “manuscript”.

All figures were revised and fixed through “Preflight Analysis and Conversion Engine (PACE) digital diagnostic tool”, as required.

We hope this new version meets the standards of PlosOne as well as the requirements of the editor and the reviewers. 

-Thank you very much for your suggestion, we revised this aspect of the MS and we hope that now it meets the formatting standards of PLOSOne.

2. Please take this opportunity to be sure you have met all of our guidelines for new species. For proper registration of a new zoological taxon, we require two specific statements to be included in your manuscript.

a) In the Results section, the globally unique identifier (GUID), currently in the form of a Life Science Identifier (LSID), should be listed under the new species name, for example:

Anochetus boltoni Fisher sp. nov. urn:lsid:zoobank.org:act:B6C072CF-1CA6-40C7-8396-534E91EF7FBB

-Thank you very much for your suggestion, we have now included LSID of the species in the results section.

Another LSID for the manuscript itself should also appear within the Nomenclature statement. You will need to contact Zoobank (zoobank.org/About) to obtain a GUID (LSID). You should receive one LSID for your manuscript and a separate, unique LSID for the new species. 

b) Please also insert the following text into the Methods section, in a sub-section to be called "Nomenclatural Acts":

The electronic edition of this article conforms to the requirements of the amended International Code of Zoological Nomenclature, and hence the new names contained herein are available under that Code from the electronic edition of this article. This published work and the nomenclatural acts it contains have been registered in ZooBank, the online registration system for the ICZN. The ZooBank LSIDs (Life Science Identifiers) can be resolved and the associated information viewed through any standard web browser by appending the LSID to the prefix "http://zoobank.org/". The LSID for this publication is: urn:lsid:zoobank.org:pub: XXXXXXX. The electronic edition of this work was published in a journal with an ISSN, and has been archived and is available from the following digital repositories: PubMed Central, LOCKSS [author to insert any additional repositories].

-Thank you very much for your suggestion. We included and completed this statement and information in the MS.

All PLOS ONE articles are deposited in PubMed Central and LOCKSS. If your institute, or those of your co-authors, has its own repository, we recommend that you also deposit the published online article there and include the name in your article.

Following a recent ruling by the International Commission on Zoological Nomenclature, electronic journals are now a valid format for publication of new zoological taxa. In order to ensure the valid publication of your new species, please be sure to include the updated version of Nomenclatural Acts (above). A complete explanation of our guidelines for publishing new species can be found on our website: http://www.plosone.org/static/guidelines#zoological.

-Thank you very much for your suggestion, we included this information in the article.

3. Please expand the acronym “MINEDUC” (as indicated in your financial disclosure) so that it states the name of your funders in full.

-“MINEDUC” is the Spanish acronym of Ministry of Education of Chile, and is already expanded in the funding section. Should we also include this information in the cover letter? If that is the case please let us know this.

"We are grateful to the community of Estero Derecho Private Protected Area and Natural Sanctuary, and to esteroderecho.cl (Coquimbo Region, Chile) for providing field support and logistics access to their plots. We are also grateful to Juan Calderón (ULS) and Alberto Castex (fundacion-indomita.org) for providing field logistics support, and to Francisco A. Squeo and Lutgarda “Lutty” Arriagada (ULS-IEB) for helping to manage the field work in Estero Derecho Private Protected Area and Natural Sanctuary. We also thank Gerardo Vergara (INFOR) for managing the financing of the project Integrated Forest Ecosystem Assessment and Monitoring System (SIMEF by its Spanish acronym). AAOA thanks the PICT 2019-597 project by Agencia Nacional de Promoción Científica y Tecnológica (Argentina). JPA thanks the DIDULS PR2121210 and DIDULS PEQMEN212124 projects of the University of La Serena, Chile, and the funding from the Ministry of Education of Chile, through MINEDUC’s performance agreement: Implementation of a competitive model of innovation and creation: preparing the University of La Serena for 2030, ULS19101, ANID FB210006 grant, and the SIMEF project (INFOR-IEB Agreement)."

 "DIDULS PR2121210 and DIDULS PEQMEN212124 projects of the University of La Serena, Chile, and the funding from the Ministry of Education of Chile, through MINEDUC’s performance agreement: Implementation of a competitive model of innovation and creation: preparing the University of La Serena for 2030, ULS19101, ANID FB210006 grant, and the SIMEF project (INFOR-IEB Agreement) to JPA.

PICT 2019-597 project by Agencia Nacional de Promoción Científica y Tecnológica (Argentina) to AAOA."

-Thank you very much for your observation. We removed all fundings from the acknowledgements section. Funding information in both parts of the text was actually the same but only presented differently, therefore we consider that there is no need to change the funding statement.

5. We note that Figures 1 and 2 in your submission contain [map/satellite] images which may be copyrighted. All PLOS content is published under the Creative Commons Attribution License (CC BY 4.0), which means that the manuscript, images, and Supporting Information files will be freely available online, and any third party is permitted to access, download, copy, distribute, and use these materials in any way, even commercially, with proper attribution. For these reasons, we cannot publish previously copyrighted maps or satellite images created using proprietary data, such as Google software (Google Maps, Street View, and Earth). For more information, see our copyright guidelines: http://journals.plos.org/plosone/s/licenses-and-copyright.

a) You may seek permission from the original copyright holder of Figures 1 and 2 to publish the content specifically under the CC BY 4.0 license. 

-Thank you very much for your comment. 

We have now generated a new figure 1 using USGS National Map Viewer (http://viewer.nationalmap.gov/viewer/), which is one of the sources suggested by PlosOne. We have included this information in the new version of the MS.

Figure 2 only presents photos taken by our team, with digital cameras or digital cameras attached to drones, we have now clearly stated this in the revised version of the MS.

Reviewers' comments:

Reviewer's Responses to Questions

Comments to the Author

1. Is the manuscript technically sound, and do the data support the conclusions?

Reviewer #1: Yes

Reviewer #2: Yes

2. Has the statistical analysis been performed appropriately and rigorously?

Reviewer #1: Yes

Reviewer #2: Yes

3. Have the authors made all data underlying the findings in their manuscript fully available?

Reviewer #1: Yes

Reviewer #2: Yes

4. Is the manuscript presented in an intelligible fashion and written in standard English?

Reviewer #1: Yes

Reviewer #2: Yes

5. Review Comments to the Author

Reviewer #1: The manuscript submitted describes a new cryptic species of Bothriurus from an unexplored locality in the Chilean Andes. The description is comprehensive and includes high-quality pictures, including photos under UV light, and photos of the male hemispermatophore. I have a few suggestions that could be easily fixed. One potential relevant issue is to incorporate an explanation on how the landmarks where defined in the segment V of both species (e.g., using carinae, macrosetae, relative position etc.). This way other authors could replicate the landmarks across additional species in further contributions.

-Thank you very much for your comments. We agree with your suggestion and we have included in the Methods section an explanation of how each landmark was taken.

Reviewer #2: The manuscript “Description of Bothriurus mistral n. sp., the highest-dwelling Bothriurus from the western Andes (Scorpiones, Bothriuridae), using multiple morphometric approaches” represents an important and interesting contribution to the knowledge of the diversity of scorpions in the region of the north-central Andes. The study not only provides the highest elevational discovery for Bothriurus in the western slopes of the Andes, but also uses the geometric morphometrics analyses for the first time to delimit a recent species within this order of arachnids. It is a great pity that the authors failed to add at least the basic characteristics and variability of genetic markers to the morphometric analyzes of the studied species (B. mistral n. sp. and B. corciareus). In that case, the results would have an even better informative value.

It is a relatively short manuscript that is carefully prepared. I have only a few comments about it, which can contribute to a better use of the presented data in possible subsequent studies.

The results represent interesting original data. In my opinion, this manuscript should be accepted for publication in Plos One after correction of some details.

Comments/recommendations:

Material and Methods

Studied material

Lines 111-119: Please add GPS coordinates also to the material of Bothriurus coriaceus.

-Thank you very much for your suggestion, we included these data in the revised version of the manuscript. 

Nomenclatorial acts

Lines 146-156: I do not understand this whole long passage. In my opinion, it is clear that the code must be followed and that the name is registered in ZooBank. This entire paragraph can only be replaced by the specified internet link to the registered name.

-We understand your concern, but this paragraph is added because policies of PlosOne demand authors to include it.

Geometric morphometric approach

Line 175: Seventeen landmarks are mentioned (Fig. 3). However, 20 points are shown in the Fig. 3. It is not clear; how did you define landmark 13 and 14. The position is clearly different between species.

-Thank you very much for your observations. We used 20 landmarks, the number 17 is our mistake, and we fixed it in the new manuscript. Landmarks 13 and 14 correspond to lateral macrosetae. We explained each landmark in the new version of the MS also following the suggestion of the other reviewer. 

Results

Line 203: Specify the registration collection number of holotype.

-Done

Lines 205-206: What does it mean: “J. Pizarro-Araya, F.M. Alfaro, J. Calderón, A. Castex (MNHN)”. All mentioned persons are owners of the collection or collected (leg.) the samples? Please, clarify this information (in the whole Type material).

-Thank you very much for your comment, these are the collectors, we specified it in the revised version of the MS. 

Line 206: What does it mean: “Conglomerate no. 41141“. It seems to like registration collection number. In this case I don’t understand that this material may be located in different collections (SIMEF-Project, MACN, MZUC, LEULS). Please, clarify this information and add the real registration collection numbers.

-Thank you very much for your comment; the term “conglomerate” refers to collection sites as defined by the standard techniques in SIMEF campaigns. We clarified this point in the revised version of the MS. 

Lines 229-230: Specify separately the measurements for holotype and paratypes (include separately this information also in Table 1). The values of Total length from the text do not correspond to the data in Table 1.

-Thank you very much for your suggestion, we included the data of male holotype in table 1, and separated it from paratypes. We also fixed the problem with the inconsistencies in the measurements.

Line 234: The holotype is only one individual (male in this case). Please correct the legend.

-Done

Table 1. – Here is no variability of the measurements. The morphometric information seems to be important for the characteristic of species. Please present the minimal, maximal, and mean values for all measurements. And separately for holotype and all paratypes.

-Thank you very much for your suggestion. We included in the description the variability in several characters which are commonly used in Bothriurid taxonomy (including maximal, minimal and mean values). Additionally we included in the revised version of the manuscript complete measurements of the holotype male, three male paratypes and three female paratypes, therefore presenting the complete measurements of four males specimens and three female specimens, which is in line with most publications in scorpion taxonomy. It is not possible for us to measure the whole group of specimens because most of the material has already been split between collections in two different countries.

Line 238: Duplication of Line 234.

-Fixed

Line 240-278: More then one page of color description seems to be very precise description of this characteristic. However, the coloration may be variable, and it is difficult to navigate in a long text. Moreover, the Figs 2A and 4A–4D are not informative in this case. The pigment pattern is used as important diagnostic characteristics. It is a reason I fully recommended to add more detailed pictures (photographs or drawing) of described coloration (with visible pattern, not as in Figs 2A and 4A–4D). it would be best to also add comparison with B. coriaceus.

-Thank you very much for your suggestions. We understand that the color description is a bit long, but it follows current standards in Bothriurid taxonomy. We agree with the reviewer that the diagnostic characters of color are not well depicted in the presented version of the manuscript; therefore we added the suggested figure of diagnostic characters of B. mistral n. sp. We also added the requested photos of B. coriaceus, as suggested, to make possible a comparison between species (all this in new figure 8). 

Line 397: Terms “shorter and wider“ are quite confusing. Please add specific values in both species.

-Thank you very much for your suggestion. We agree with this point and -included these data in the revised version of the MS.

Please, control the formal style of Figure references all over the text. It is mix of “Fig” and Fig.” Moreover “Figure” is not abbreviated in the figure legends. I assume that the formatting of the text will be subject to an even more detailed check by the editorial team.

-Done

6. PLOS authors have the option to publish the peer review history of their article (what does this mean?). If published, this will include your full peer review and any attached files.

Do you want your identity to be public for this peer review? For information about this choice, including consent withdrawal, please see our Privacy Policy.

Reviewer #1: Yes: Jairo A. Moreno-González

Reviewer #2: No

-Done

---

## [Decision Letter · Decision Letter 1]

23 Jan 2023

Description of Bothriurus mistral n. sp., the highest-dwelling Bothriurus from the western Andes (Scorpiones, Bothriuridae), using multiple morphometric approaches

PONE-D-22-27368R1

Dear Dr. Ojanguren-Affilastro,

We’re pleased to inform you that your manuscript has been judged scientifically suitable for publication and will be formally accepted for publication once it meets all outstanding technical requirements.

Kind regards,

Michael Schubert

Academic Editor

PLOS ONE

Reviewers' comments:

Reviewer's Responses to Questions

**Comments to the Author**

1. If the authors have adequately addressed your comments raised in a previous round of review and you feel that this manuscript is now acceptable for publication, you may indicate that here to bypass the “Comments to the Author” section, enter your conflict of interest statement in the “Confidential to Editor” section, and submit your "Accept" recommendation.

Reviewer #1: All comments have been addressed

Reviewer #2: All comments have been addressed

2. Is the manuscript technically sound, and do the data support the conclusions?

Reviewer #1: Yes

Reviewer #2: Yes

3. Has the statistical analysis been performed appropriately and rigorously? 

Reviewer #1: (No Response)

Reviewer #2: Yes

4. Have the authors made all data underlying the findings in their manuscript fully available?

Reviewer #1: Yes

Reviewer #2: Yes

5. Is the manuscript presented in an intelligible fashion and written in standard English?

Reviewer #1: Yes

Reviewer #2: Yes

6. Review Comments to the Author

Reviewer #1: (No Response)

Reviewer #2: The authors have carefully edited the manuscript based on all the comments and recommendations. For this reason, the article can now be fully supported for acceptance and publication in the journal PLOS ONE.

7. PLOS authors have the option to publish the peer review history of their article (what does this mean?). If published, this will include your full peer review and any attached files.

Reviewer #1: **Yes: **Jairo A. Moreno-Gonzalez

Reviewer #2: No

---

## [Editor Report · Acceptance letter]

26 Jan 2023

PONE-D-22-27368R1 

Description of *Bothriurus* mistral n. sp., the highest-dwelling *Bothriurus* from the western Andes (Scorpiones, Bothriuridae), using multiple morphometric approaches 

Dear Dr. Ojanguren-Affilastro:

I'm pleased to inform you that your manuscript has been deemed suitable for publication in PLOS ONE. Congratulations! Your manuscript is now with our production department. 

Kind regards, 

on behalf of

Dr. Michael Schubert 

Academic Editor

PLOS ONE